# Position: The AI Conference Peer Review Crisis Demands Author Feedback and Reviewer Rewards

Jaeho Kim [* 1]   Yunseok Lee [* 1]   Seulki Lee [1]

## Abstract

The peer review process in major artificial intelligence (AI) conferences faces unprecedented challenges with the surge of paper submissions (exceeding 10,000 submissions per venue), accompanied by growing concerns over review quality and reviewer responsibility. This position paper argues for **the need to transform the traditional one-way review system into a bi-directional feedback loop where authors evaluate review quality and reviewers earn formal accreditation, creating an accountability framework that promotes a sustainable, high-quality peer review system.** The current review system can be viewed as an interaction between three parties: the authors, reviewers, and system (*i.e.* conference), where we posit that all three parties share responsibility for the current problems. However, issues with authors can only be addressed through policy enforcement and detection tools, and ethical concerns can only be corrected through self-reflection. As such, this paper focuses on reforming reviewer accountability with systematic rewards through two key mechanisms: (1) a two-stage bi-directional review system that allows authors to evaluate reviews while minimizing retaliatory behavior, (2) a systematic reviewer reward system that incentivizes quality reviewing. We ask for the community's strong interest in these problems and the reforms that are needed to enhance the peer review process.

## 1. Introduction

Artificial intelligence (AI) conferences (*e.g.* ICML, ICLR, NeurIPS, CVPR, etc) play a crucial role as a premier venue

---
[*]Equal contribution [1]Artificial Intelligence Graduate School, Ulsan National Institute of Science and Technology (UNIST), Ulsan, South Korea. Correspondence to: Seulki Lee <seulki.lee@unist.ac.kr>.

*Proceedings of the 42nd International Conference on Machine Learning*, Vancouver, Canada. PMLR 267, 2025. Copyright 2025 by the author(s).

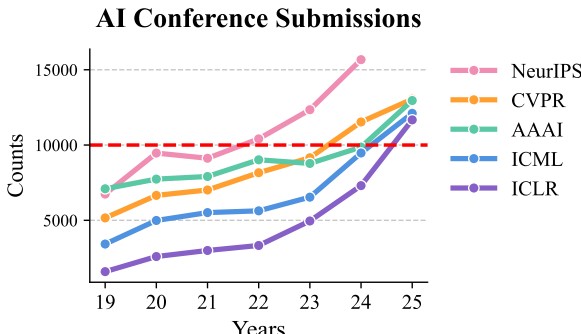

**Figure 1: AI Conference Submission Counts.** The number of paper submissions to most major AI conferences (*e.g.* NeurIPS, CVPR, AAAI, ICML, ICLR) exceeded 10,000 by 2025. For example, there was a 59.8% increase in ICLR submissions in 2025 alone. We forecast similar growth in other venues as well.

for cutting-edge research dissemination, fostering intellectual discourse and collaboration among researchers. Unlike traditional scientific areas where journals are the primary publication outlets (Kim, 2019), the fast-paced nature of AI research has elevated these conferences to the status of first-tier venues, with their double-blind peer-reviewed papers carrying impact comparable to many prestigious journals (Freyne et al., 2010). However, over recent years, the number of paper submissions to major AI conferences has skyrocketed (Figure 1), overwhelming the traditional peer review system (Tran et al., 2020). **With such rapid growth, the authors believe that the current AI conference system is becoming increasingly unsustainable where the hard-earned conference's established reputation is breaking down.** To put this more directly, this translates to *declined review quality.*[1] This issue, however, cannot be attributed solely to the reviewers - the responsibility and consequent adverse effects are shared among all three main stakeholders: Authors, Reviewers, and the System.

**Definition 1.1** (**Authors**). Authors are individuals who submit manuscripts to AI conferences for peer review and po-

---
[1]We stress that the majority of authors and reviewers maintain high professional standards. However, even a small number of participants who deviate from these standards can significantly impact the reputation and trust built on these peer review process, creating challenges that affect the entire AI community.

tential publication.

**Definition 1.2** (**Reviewers**). Reviewers are qualified experts, selected from the author community or invited based on expertise, who evaluate submitted manuscripts.

**Definition 1.3** (**System**). System includes the venue (*e.g.* ICML) and platforms (*i.e.* OpenReview [2]) that supervise the overall peer review process and AI conferences.

Throughout this position paper, we outline why the responsibility of *declined review quality* is shared by all three main stakeholders and categorize the challenges as either addressable or non-addressable. We then propose our minimal yet actionable approach that addresses the peer review challenges, positing that just as authors are incentivized to produce high-quality papers, reviewers should be similarly motivated through systematic rewards for quality reviews. Specifically, we advocate for

1. **Author Feedback:** A two-stage double-blind peer review system where authors evaluate review quality through objective criteria, with safeguards protecting reviewers from potential retaliation when providing critical reviews.
2. **Reviewer Rewards:** A system-backed reward framework that provides incentives for thorough reviews, where reviewers can accumulate and showcase their review contributions as verifiable academic credentials, creating long-term professional value.

Our paper is structured as follows. Section 2 discusses the root causes of the current problems in AI conference peer reviews. Section 3 suggests our minimal, yet actionable changes that can be made to the current peer review process to restructure the power balance between reviewers and authors. Then, Section 4 provides concrete reward systems for reviewers which need the support of the Systems. Consequently, Section 5 elaborates on the steps needed to make these changes happen and discusses practical challenges of our proposals. Section 6 discusses alternative views from our position. Section 7 includes related works to ours. Lastly, Section 8 concludes our position with future directions to improve the peer review process in AI conferences.

## 2. Reasons for Degraded Review Quality

This section looks into some of the direct causes that negatively affect the review quality of the current AI conference. We intentionally defer discussions on related but important problems in the peer review process of AI conferences such as collusion rings (Jecmen et al., 2024), infringements of the double-blind process (Russo, 2021), and others (Shah, 2022; Centeno et al., 2015; Gao et al., 2019; Ross et al., 2006),

---

[2]https://openreview.net/

as these require different approaches (*e.g.* through policy enforcement and detection tools) from the review quality issues that we address in this paper.

### 2.1. Non-Addressable Causes

**Increase of Paper Submission.** The surge in paper submissions to major AI conferences (Huang et al., 2023) in recent years is undoubtedly the main reason for the decline in review quality, a challenge caused by authors in the three main stakeholders (authors, reviewers, and the system) of the peer review process. The recent interest (or frenzy) in AI across academics, governments, and industrial sectors (Maslej et al., 2024) has pushed unprecedented growth in AI research participation. This growth has created a cycle where securing and maintaining research funding often requires frequent top-tier publications (Stephan, 2012), driving up submission numbers across AI conferences.

This surge is further intensified by technological and academic factors in the research community. On the technological front, the use of large language models (LLMs) lowers the barrier to paper writing, with claims that more than 17.5% of computer science papers have been modified or produced with LLMs just in 2024 (Liang et al.). This potentially enables authors to participate in multiple submissions. On the academic side, the "publish or perish" culture (Van Dalen & Henkens, 2012) creates pressure for frequent publications. This pressure leads to researchers prioritizing quantity over quality, leading to a strategic submission of incomplete work to obtain peer reviews (*i.e.* review shopping) and use these reviews for improvement, placing additional burdens on the reviewers.

**Momentum in AI.** The rapid expansion of AI, both in scale and diversity, means that novel research topics and methodologies appear every year. We found that on average 17% of top 20 research keywords in a major AI conference change annually (Appendix A), indicating the fast shift in research trends. This makes it challenging for reviewers to stay current with new methodologies, even for those working in the specific field. Unlike the partially addressable issues of matching papers with appropriate reviewers through policy advancement (Wu et al., 2021), this challenge stems from the inevitable knowledge gap that affects review quality.

We consider these as the *Author*-side causes, which cannot be addressed by improving the peer review process. This leads to what we term a *review quality ceiling*, where even the most perfect implementation of the review process cannot overcome these challenges that degrade review quality.

### 2.2. Addressable Issues

**Reviewer Negligence: A Power Imbalance.** The peer review process inherently places authors in a vulnerable position, while positioning reviewers on the advantageous

side of a power imbalance. Simply put, when a paper is rejected (regardless of its academic contribution), the authors suffer the consequences, while reviewers face no accountability. In such imbalanced circumstances, some reviewers neglect their responsibilities knowing that their reviews will not affect their careers. This can lead to a superficial review of the paper's content, failing to engage with its core contributions and provide constructive feedback to help authors improve their work. Another recent issue is the use of LLMs to generate peer reviews. Recent works (Liang et al., 2024a) have highlighted that the use of certain adjectives such as 'commendable' and 'innovative' has increased significantly in ICLR peer reviews since the advent of ChatGPT, indicating the possible use of LLMs for peer reviews. Our evaluation (Appendix B) also shows a significant increase in the ratio of reviews without any typos, indirectly indicating the wide usage of LLMs to refine or generate these reviews[3]. Unfortunately, at the current level of technology, LLM-generated reviews lack technical depth and fail to provide constructive feedback like human reviewers (Zhou et al., 2024; Ye et al., 2024). Also, AI detection tools are still not reliable (Elkhatat et al., 2023), making it hard for authors to definitively identify and challenge such reviews.

We view these as the *Reviewer*-side causes, which could potentially be addressed by reconsidering the power balance between authors and reviewers. We are not advocating for a balance of 50-50, but there should be minimal protocols to ensure reviewers maintain a basic standard of responsibility in their evaluations and accountability over their work.

**Reviewer Exploitation.** The current peer review system neither holds reviewers accountable nor adequately rewards their efforts. While the lack of recognition or incentives doesn't necessarily compromise review quality, it does little to enhance it, offering minimal motivation for reviewers. The perspective that peer review should be an academic voluntary service (Northcraft & Tenbrunsel, 2011), where intellectual curiosity and scholarly responsibility are expected to be sufficient drivers, overlooks the realities of the immense workload of reviewers in AI conferences. The burden is intensified by short review deadlines and the dual responsibility many face as both authors and reviewers in the same conference submission.[4] In such a demanding scenario, it becomes difficult to request reviewers to enhance or maintain consistent review quality without any systematic rewards or incentives for their efforts. Although some conferences recognize top reviewers through awards and public acknowledgments, these awards or recognition are not easily visible enough to provide meaningful motivation to the broader reviewer community.

---

[3]We acknowledge that there exists confounding factors (*e.g.* the use of correction tools)

[4]Several conference requires authors to review a certain number of papers as part of their submission process.

We argue that this is a *System*-side cause, where the System has fallen short in motivating reviewers through meaningful professional recognition and rewards, relying instead on a traditional model of voluntary academic service. Such a model appears insufficient for meeting the demands of the current AI conference and, as such, a System-backed reward system is necessary to motivate current and future peer reviewers and praise for their quality reviews.

## 3. Redesigning Peer Review: A Two-way Method

To address the power imbalance between reviewers and authors, we propose a bi-directional, two-stage review method. Our proposed approach represents a minimal modification to the current double-blind peer review system widely adopted by AI conference venues, making it a low-risk intervention. Importantly, our system works within existing review timelines, ensuring practical implementation while addressing reviewer-author power imbalance issues. As such, we provide a brief description of the current double-blind peer review process and illustrate the changes that we propose.

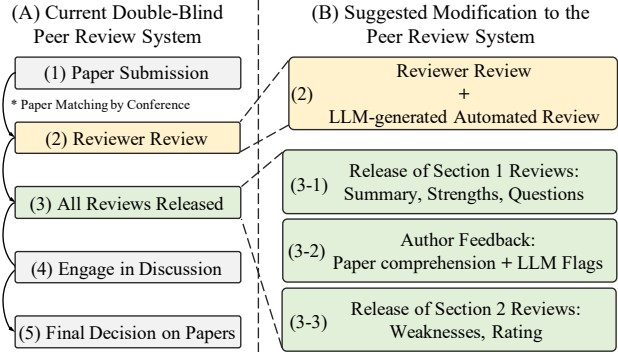

Figure 2: **Suggested Modification to the Peer Review System.** (A) Overview of the standard double-blind peer review system currently adopted by most academic conferences. (B) Our proposed modification to steps two and three of the existing peer review system. Our modification is minimal and it does not disrupts existing timelines, making it easily adaptable to existing systems.

**Current Review System.** The paper submission and decision process in major AI conferences (*e.g.* ICML, NeurIPS) follows the steps illustrated in Figure 2. The process proceeds as follows: (1) Authors submit their papers to the the System (*e.g.* OpenReview) and papers are matched with reviewers based on the System's policy (*e.g.* paper bidding, research expertise); (2) Reviewers evaluate their assigned papers and submit their reviews; (3) All reviews and ratings are simultaneously released to authors; (4) Authors and reviewers engage in discussion to address potential misunderstandings, during which reviewers may adjust their scores based on clarifications; and (5) Meta-reviewers (*i.e.* area chairs) make final decisions based on both the manuscript

and its reviews. This is a high-level overview where details may vary across conferences (Kuznetsov et al., 2024).

**1. Addressing LLM Reviews.** To first address LLM reviews, we propose to incorporate LLM-generated peer reviews alongside human reviews during the second stage when reviewers evaluate their assigned papers. This could potentially discourage reviewers from solely relying on LLMs for peer review, as similar AI-generated content would be visible to authors. Importantly, this LLM review would be exclusively visible to authors, who would be explicitly informed of their origin and are not required to engage in discussion with these reviews. While reviewers would be notified about the inclusion of LLM-generated reviews in the process, details about the type of LLM service (*e.g.* ChatGPT, Claude) and prompts would remain confidential. The inclusion of LLM-generated reviews serves two main purposes: (1) LLM reviews act as a psychological deterrent against the few irresponsible reviewers who might otherwise rely entirely on LLMs for evaluations, as they know the System is already incorporating such automated reviews, and (2) it provides authors with a soft reference point to identify and flag potential LLM-generated reviews, detailed in our second proposal.

**2. Two-Stage Release of Reviews.** We propose a sequential release of review content rather than the conventional simultaneous disclosure of all reviews and ratings. Specifically, we divide review content into two distinct sections: section one includes neutral to positive elements, including paper summary, strengths, and clarifying questions, while section two contains more critical parts such as weaknesses and overall ratings. Between these releases, authors evaluate reviews from section one based on two key criteria: (1) the reviewer's comprehension of the paper and (2) the constructiveness of questions in demonstrating careful reading. During this feedback phase, authors can also flag potential LLM-generated reviews[5] by comparing them with the provided LLM reviews. This two-stage disclosure prevents retaliatory scoring while providing the minimal safeguards necessary for a fair review. Once authors complete their feedback, section two is promptly disclosed, and the authors are not allowed to modify their evaluations. Subsequently, the review process follows the conventional workflow.

The author feedback scores on reviews function in two ways. **In the long term,** these scores can be used for the reviewer reward system, detailed in Section 4. **In the short term,** the feedback scores are used in the final meta-review stage as additional context in assessing the reviews made on the paper. Specifically, meta-reviewers have access to both the authors' feedback score of the paper being evaluated and

the averaged feedback scores across multiple submissions conducted by the reviewer. Since reviewers typically evaluate multiple papers, the aggregated feedbacks made on the reviewer by the authors provide a reliable measure of review quality while minimizing the impact of potential retaliatory scores from (some malicious) authors. Similarly, if LLM flags are constantly raised against a specific reviewer across multiple submissions, this can alert meta-reviewers to take appropriate actions to maintain review quality. Moreover, consistently low average feedback scores and repeated LLM flags could lead to restrictions on future reviewer appointments and manuscript submissions. The need for such regulations has already been recognized and implemented in the broader AI community. For instance, CVPR 2025 has independently implemented a similar policy[6], where 19 papers authored by reviewers flagged as "highly irresponsible" by the meta-reviewer were desk-rejected that might otherwise have been accepted. While meta-reviewer judgment could be reliable, our proposed system provides supplementary quantitative metrics of aggregated feedback scores and LLM flags from authors, offering additional context to support the decision-making process.

Our proposal comes with two key advantages. First, our suggestion is a manageable change that can be integrated into the existing peer review process of AI conferences, without compromising the current timelines. Making sudden changes to the peer review system might face resistance or implementation challenges, whereas our proposal builds upon conventional workflows. Importantly, our suggested modifications maintain the efficiency of existing processes: LLM reviews can be generated while other human reviewers make their evaluations, and the sequential release of reviews does not interrupt the discussion phase. Second, our proposal provides a basic safeguard for both authors and reviewers, where authors are protected from unprofessional reviewers, and reviewers are protected from retaliatory scoring by authors. The objective of author feedback on the reviews is not to impose additional burdens on the reviewers but rather to motivate reviewers to write constructive feedback and maintain the review quality standards that the AI community should uphold.

We acknowledge several practical concerns that can be made with our suggestion. First, one might worry that reviewers may focus solely on writing overly lengthy strengths to avoid negative author feedback, while providing less constructive feedback to improve the paper. To address this, there is a need for a length limitation, to make reviews remain concise. Moreover, being able to write the proper strengths of a submission requires a thorough understanding of both the paper and its related fields. A low-value review

---

[5]LLMs are useful tools for refining writing, but we oppose the use of LLMs to generate reviews entirely, as the technical depth and reviews made with current LLM cannot match human reviewers' expertise in evaluating novel contributions.

[6]https://cvpr.thecvf.com/Conferences/2025/CVPRChanges

that simply lists positive aspects would likely receive low feedback scores from authors. Second, there might be submissions that are incomplete or lack substantial academic value, making it hard for reviewers to identify meaningful strengths. However, such papers should typically be filtered out during the desk rejection phase, or would receive low ratings across multiple reviewers. In these cases, the System can be designed to exempt reviewers from author feedback, protecting them from unfair evaluations. Third, LLM flags could be misused by authors attempting to discredit valid reviews. However, meta-reviewer's oversight can alleviate such problems and the System could require authors to provide specific justification when raising them.

## 4. A Concrete Reward System

Several AI conferences have implemented recognition programs to acknowledge reviewer contributions. As of 2024, ten out of fourteen major AI conferences publicly recognize outstanding reviewers through their Reviewer Hall of Fame programs on their respective websites, with a subset offering in-kind compensation to top-performing reviewers (detailed in Appendix C). Although reviewers can showcase these recognitions on their personal platform (*e.g.* websites), the current reward system remains largely unnoticeable to the broader academic community and lacks professional impact beyond official documentation of their service (Warne, 2016). As such, **we argue that the reviewer reward system should become more visible and accessible to the general public and academia in order to better motivate reviewers, and as a result enhance the overall review quality in AI conferences.** Ultimately, reviewers' contributions should carry long-term professional academic value, as they contribute to the production and dissemination of high-quality research papers (Mulligan et al., 2013).

### 4.1. Digital Badge System

To make reviewer achievements more visible and provide motivation for reviewers, we propose the use of digital badge systems (Gibson et al., 2015). Digital badges work as a digital credential and have been helpful in the education sector to promote motivation, engagement, and achievement of learners (Facey-Shaw et al., 2017). Evidence from a higher education study suggests that the *incorporation of badges used as peer review has increased participation and "richer comments" on the discussion boards along with an enhanced sense of community* (Stefaniak & Carey, 2019). Building on these findings, we propose a badge system for peer reviewers in AI conferences, where digital badges can be displayed on academic profiles (*e.g.* OpenReview Profile, Google Scholar Profile) as shown in Figure 3. Badges are officially issued by conferences but should follow a standardized design and implementation policy between venues.

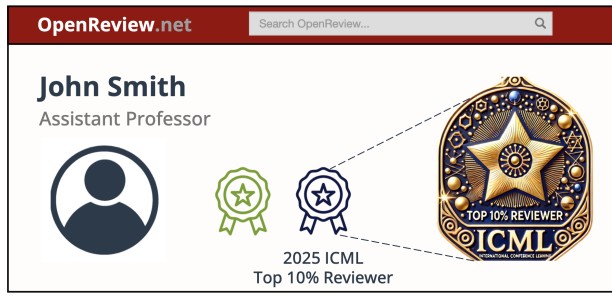

**Figure 3: Reviewer Digital Badge System.** We provide an illustrative figure on how digital badges could be displayed on academic profiles (*e.g.* OpenReview). Digital badges are officially issued by the conferences and reviewers can obtain these badges through our proposed author feedback scores or by meeting specific criteria established by the conference venues.

Specifically, we recommend implementing top 10% and 30% reviewer badges, as excessive diversity of badges could become overwhelming and reduce their effectiveness. Here, the author feedback scores on the reviews proposed in Section 3 can be used as quantitative measures for determining badge eligibility. In the long term, these badges will serve as meaningful academic credentials and statistics for research groups, fostering a culture where high-quality peer review is better recognized and valued. This recognition can extend beyond mere acknowledgment, as conferences can use these badges as objective criteria when selecting leadership roles like area chairs and senior program members.

### 4.2. Tracking Reviewer Impact

While badges provide visible recognition, we need systematic ways to measure and track reviewers' contributions over time. To address this need, we propose two measures to track and quantify reviewer impact in AI conferences. First, we propose a reviewer activity tracking system that documents reviewers' contributions to conferences. While several journal publishers already utilize such systems (*e.g.* Publons, ReviewerCredits) where reviewers can optionally disclose their review activities and content, most reviewer feedback in OpenReview remains anonymous to the public. This anonymity makes it difficult for reviewers to receive recognition for their efforts. Extending such systems to AI conferences could increase reviewer motivation by making their contributions more visible and quantifiable. Second, we stress the need for a reviewer impact score, a metric to quantify the impacts of the reviewed papers by the reviewers. Similar to how the h-index measures an author's research impact (Bornmann & Daniel, 2007), this metric treats reviewers as authors of the papers they have reviewed, evaluating their contribution based on the subsequent impact of these works. Currently, most scholarly metrics (*e.g.* citations, h-index, i10-index) focus on the impact that is centered on the author's research activity, while the con-

tributions of reviewers to the scientific literature remain less quantified. This metric provides a quantifiable measure of reviewer impact and motivates reviewers to identify high-quality research papers. Both proposals are technically feasible to implement, as many major AI conferences now use OpenReview as their primary platform for peer review (Huang et al., 2023). In the long term, implementing these systems would help researchers build comprehensive academic portfolios that reflect both their research contributions and their service to the community.

### 4.3. Other Rewards

A survey on journal peer reviewers indicates that in-kind compensation (*e.g.* free subscriptions, publication charge waivers) most effectively motivates reviewers to continue their service (Mulligan et al., 2013). However, among the major AI conferences we examined, only four offered any form of compensation to reviewers, typically through conference registration fee waivers. This limited financial support is understandable given the budget limits of academic conferences. As such, conferences should consider cost-effective approaches to provide tangible rewards to reviewers. This can be done through implementing symbolic recognition such as specially marked conference name tags and merchandise for top reviewers. For instance, CVPR has provided dedicated poster stickers to top reviewers in 2024. While these rewards may seem modest, they serve as visible acknowledgment and could foster a culture of respect and recognition for peer review contributions within the research community. Beyond physical tokens, conferences can organize educational workshops that can benefit both reviewers and the broader academic community. Reviewers can share their reviewing practices, which could potentially benefit authors as well. These sessions could be documented and published as conference proceedings, creating citable contributions for reviewers.

## 5. Discussions

### 5.1. Need for Gradual Implementations

While we believe that our proposal is a minimal and feasible change that is needed to make AI conferences more sustainable, not everyone may feel this way. In this sense, we suggest the following two steps that are needed for the gradual adoption of our proposals. First, a large-scale survey from the authors and reviewers regarding the current peer review system in AI conferences is necessary. The survey should be designed to reflect the different perspectives and responsibilities of each role in the review process, including peer reviewers and meta-reviewers. Understanding the distinct challenges faced by reviewers at different levels will provide a comprehensive understanding of how the peer review system can be improved. Second, we sug-

gest implementing our proposed modification to the review system as a pilot program in smaller-scale tracks or workshops before considering broader adoption. Implementing them in a smaller-scale peer review process could provide practical insights into the feasibility of our proposals, while minimizing disruption to the existing review process.

### 5.2. Practical Challenges

The most practical challenge in implementing our proposed methods is to elicit existing Systems (*e.g.* venues, OpenReview) to make these changes happen. In particular, the reviewer feedback and reviewer reward systems cannot be initiated without cooperation from multiple venues and OpenReview developers. Making changes to the existing peer review system, which appears to work well on the surface, would require immense effort from multiple parties.

In addition, the question of who would be responsible for implementing these changes' financial costs needs to be answered. Multiple venues could create a collective fund for implementing these changes, but even very big venues often face budgetary constraints. For instance, ICML 2024 initially planned to provide in-kind compensation to the top 10% of reviewers but had to reduce this to a smaller pool of top reviewers due to financial constraints[7]. These practical concerns highlight the challenges of implementing the proposed changes to the current peer review process.

### 5.3. Potential Gaming by Reviewers

We recognize that while digital badges leverage positive gamification elements (*e.g.* participation, motivation), they could potentially incentivize reviewers to optimize for rewards through overly liberal reviewing rather than maintaining review standards. To address these concerns, we should use a carefully designed evaluation metric that rewards quality over leniency. Badges should be provided to reviewers who demonstrate thoroughness, identification of crucial issues, and constructive feedback, but not simply through positivity of the assessments. Ultimately, the success of our proposed badge system depends on aligning reviewer incentives with the academic community's goal of advancing knowledge through constructive peer evaluation.

### 5.4. In the Era of LLMs

As the era of LLMs came earlier than what most people expected, it seems that most AI conferences are now hurrying to adapt to this sudden change while struggling to maintain their traditional review structures. One of the notable changes we have observed in most AI conference websites is the growing guidelines regarding LLM usage in both pa-

---

[7]https://medium.com/@icml2024pc/
reviewing-at-icml-2024-a7aa81169d8c

per writing and peer reviewing. For instance, ICLR 2025 has explicitly asked authors to submit their usage of LLMs in the paper writing process. These measures indicate that conferences are only beginning to grapple with the impact of LLMs in academic workflows. Subsequently, these measures taken by conferences raise important questions about the future of peer review itself. While the topic of whether LLMs can provide technical in-depth review is debatable, our position is that current LLMs can only generate reviews that remain superficial and lack technical depth. In this context, we cannot avoid asking, *What should happen when LLMs provide better review than most human reviewers?* This question is uncomfortable as it questions the fundamental nature of existing peer review systems and, at a deeper level, challenges the utility of human intelligence, but it is also a question that we must start thinking about as members of the AI community. We believe that the discussion on this topic should promptly begin, reminding us that as much as we were not prepared for the sudden arrival of LLMs, we cannot afford to be equally unprepared for the next advancement of LLMs which could seriously disrupt the current peer review systems.

## 6. Alternative Views

As an alternative view to our concerns about AI conference sustainability, some could contend that AI conferences are highly sustainable and that bringing the proposed changes can only burden the already over-burdened reviewers. This perspective raises several key counter-arguments that provide an alternative view to our position.

First, the peer review system in AI conferences is highly sustainable and self-evolving. The number of submissions has been constantly increasing, and the fact that several conference venues are already handling more than 10,000 submissions validates the system's effectiveness. The number of submissions does not highly affect the peer review system, as the number of qualified reviewers grows proportionally with the field. Conferences also run in a well-structured and hierarchical manner, where there are distinct roles endowed to each level of reviewers and organizers. This system can easily adapt to the growing number of submissions.

Second, there is a lack of evidence that the quality of reviews in AI conferences is declining. The perception of declining review quality likely stems from a vocal minority, where this minority does not reflect the whole AI community. Authors who have received positive reviews and have had their paper accepted are less likely to voice their opinions, while those who have received critical comments are more motivated to express their dissatisfaction (Goldberg et al., 2025). This leads to a sampling bias, which can become particularly evident in social media and academic forums (Hargittai, 2020). Researchers with a strong publication record have

limited motivation to participate in public discussion about review quality, as they have fewer incentives to question such systems which have validated their work.

Third, the implementation of author feedback is likely to make reviewer recruiting difficult, placing a burden on the System. Not many reviewers would want to get their reviews rated (Nicholson & Alperin, 2016), as this would place additional pressure and potential disadvantages on their academic activity. The fear of receiving negative feedback could make reviewers write unnecessarily lengthy reviews (Goldberg et al., 2025), leading to a general decline in review quality. This could eventually create an adversarial environment between reviewers and authors. Furthermore, implementing such a system would also increase the burden on meta-reviewers who would need to monitor the reviews and the feedback scores from the authors.

While these counterarguments present important considerations, they do not fully address the fundamental challenges that AI conferences face in maintaining long-term sustainability. Although conferences have historically scaled with submission growth, this does not guarantee that the number of qualified reviewers will grow proportionally while maintaining the high review standards as when conferences were smaller. Such expansion could come with a cost of declined review quality. Moreover, the lack of evidence for declining review quality equally suggests a lack of evidence that quality standards are being maintained or enhanced. This underscores the urgent need for conferences to conduct comprehensive surveys on peer review quality, gathering systematic feedback from both authors and reviewers. While we acknowledge that implementing our proposed changes could initially make reviewer recruitment harder, we believe this can be effectively addressed through our proposed reviewer reward system, which would help attract and retain qualified reviewers. In the long term, implementing our suggestions could outweigh the short-term adaptation costs.

## 7. Related Works

Our position paper discusses methods to improve peer review quality in AI conferences. In this section, we present related works in four key subsections: studies on review quality by AI conferences, LLMs in peer reviews, reviewer rewards, and other approaches to enhance review quality.

### 7.1. Studies on Review Quality in AI Conferences

NeurIPS has performed several comprehensive studies to analyze their peer review process. In both 2014 and 2021, they performed a review consistency experiment, where 10% of papers submitted to their venue were randomly selected and had them reviewed by two independent groups of reviewers (Cortes & Lawrence, 2021; Beygelzimer et al.,

2023). Results showed that 16-23% of papers could have been either accepted or rejected based on the reviewing group, indicating a high randomness in peer review quality. Another directly related work to ours is the study performed in 2022, where NeurIPS asked authors, reviewers, and meta-reviewers to evaluate the quality of peer reviews after the whole submission process (Goldberg et al., 2025). Among the many notable results, we highlight two key insights that directly relate to our proposals: author-outcome bias and elongated review bias. The author-outcome bias indicates a higher likelihood of authors rating the review to be helpful when the reviews recommended acceptance compared to rejection. This finding motivates the need for our two-stage review release mechanism, where authors first evaluate reviews based on their positive aspects and demonstrated understanding of the paper, before seeing the final ratings and critiques. This sequential release of reviews could prevent retaliatory scoring by authors and still obtain feedbacks on the review quality. Similarly concerning, the elongated review bias notes that longer reviews are rated higher than shorter reviews even when the two reviews contain the same information. This observation highlights the need to limit the length of the reviews that are released in the first stage of our two-stage review release method.

## 7.2. LLMs in Peer Reviews

While our position paper addresses the need for an LLM review simply to deter human reviewers from solely relying on LLMs for peer review and to provide authors as a soft reference point for flagging LLM-generated reviews, we believe that the topic of using LLMs as a general peer reviewer needs to be mentioned. Given the ongoing debate in the community, we present a balanced discussion of the potential benefits and limitations of using LLMs in the peer review process. Those who are against the use of LLMs as peer reviewers focus on the limited technical depth and insights (Donker, 2023), their tendency to reiterate limitations disclosed by authors, and their vulnerability to prompt injection attacks (Ye et al., 2024). On the other hand, some argue that with sufficient engineering, LLMs can be used to desk-reject low-quality papers to reduce the burden on reviewers (Tyser et al., 2024) or even automate the whole paper writing-to-review process (Lu et al., 2024). However, most works remain rather neutral (Liu & Shah, 2023; Kuznetsov et al., 2024; Liang et al., 2024b), discussing the potential of using LLMs to enhance the peer review process but with human oversight.

## 7.3. Reviewer Rewards

Providing reviewers with incentives has been discussed in several prior works. While we have not found works discussing the use of digital badges and reviewer impact scores as in our works, there were literatures discussing the in-corporation of "markets" and "auctions" into the academic review process (Frijters & Torgler, 2019; Srinivasan & Morgenstern, 2021). Basically, reviewers can get paid with academic tokens based on the quality of reviews, where the authors rate the quality of the reviews (Ugarov, 2023). A more direct approach suggests providing actual monetary gifts (up to $1000 per reviewer) based on a structured rubric model (Sculley et al., 2018). The meta-reviewers will rate the reviewers based on the rubric, and reviewers will receive different levels of compensation. The funding can be obtained from implementing submission fees, differential pricing of conferences, and explicit sponsorship. While academic tokens and direct monetary rewards can be an idealistic solution, the risks from practical and ethical concerns outweigh their potential benefits.

## 7.4. Other Approaches to Enhance Review Quality

Game-theoretic modeling of the peer review process has been studied for improving review quality. One approach is the use of isotonic mechanism (Su, 2021; Wu et al., 2023; 2024). Assuming that authors have multiple submissions to a venue, authors can self-rank the quality of their own works, and this additional context can be used to de-noise the ranks coming from the reviewers. The mechanism is theoretically grounded: it proves that authors maximize their utility by providing truthful rankings, and these rankings can be used to calibrate reviewer scores, thus reducing noise in the review process and enhancing review quality. This method was further validated on data collected from the peer review process in ICML 2023 (Su et al., 2024). Another approach is to model the submission process with agent-based simulations (Zhang et al., 2022), suggesting that setting a typically higher acceptance bar could reduce the burden on reviewers as most reviewing loads come from the resubmission of borderline works. They also found that slightly increasing the quality of review is better than increasing the quantity of reviews, which aligns with our position.

A large number of works try to address peer review quality by implementing improved submission policies and reviewer-author matching algorithms. Specifically, implementing more fine-grained submission tracks and communicating editorial priorities could guide reviewers to enhance review quality (Rogers & Augenstein, 2020). Matching reviewers who have expertise in the submitted paper through algorithmic advancement has been extensively studied in both methodology (Mimno & McCallum, 2007) and its actual implementation in AI venues (Xu et al., 2024). We believe that these are orthogonal approaches to our position and merit continued research as complementary solutions to improving the peer review process.

## 8. Conclusion

At this moment of heightened interest in AI from both academia and the general public, AI conferences stand at the forefront of AI development, where their significance and impact are beyond measure. Their roles of improving, validating, and disseminating papers are crucial as they serve as a vital channel for conveying important information and technology to society. Throughout this position paper, we have consistently argued that their roles as prestigious and reliable AI conferences have become highly unsustainable from all three main stakeholders: authors, reviewers, and the system itself. Given the practical constraints in regulating authors, we have proposed improvements at the reviewer and system levels. Specifically, we proposed a bi-directional review system where authors are protected with minimal safeguards from unprofessional reviewers, and a systematic reviewer reward system that incentivizes reviewers for their academic service. We have also objectively discussed the potential pros and cons of our proposals. We conclude this position paper with the hope that this work can bring these issues into broader public discourse and inspire future researchers to work on these challenges.

## Acknowledgements

This work was supported by the National Research Foundation of Korea(NRF) grant funded by the Korea government(MSIT) (No.RS-2023-00277383) and Institute of Information & communications Technology Planning & Evaluation(IITP) grant funded by the Korea government(MSIT) (No.RS-2020-II201336, Artificial Intelligence Graduate School Program(UNIST)). This work was supported by the Google Cloud Research Credits program with the award Gemma 2 Academic Program.

Although we were aware of the widespread issue of declining review quality, we would not have been motivated to write this paper without the encouragement and support of Dr.Yoontae Hwang and Dr.Seokju Hahn, to whom we are deeply grateful. We also thank Gyeongho Kim, Bosung Kim, Isu Jeong, and Kyu Hwan Lee for reviewing our manuscript. We are also grateful to the anonymous reviewers for their encouragement and constructive feedback throughout the review process.

## Impact Statement

The views and proposals presented in this position paper reflect solely the thoughts and opinions of the authors, and do not represent the official stance of any affiliated institutions, organizations, or other parties.

We believe that this position paper could spark meaningful discussion on the need to reform the current double-blind peer review process used by many AI conferences, and hopefully, conferences can make bolder decisions. Our work builds upon the valuable insights shared by numerous researchers across social media platforms such as Reddit, Twitter, and LinkedIn, whose contributions we deeply appreciate and hope to formally acknowledge in the future. This paper is an effort to bring these productive discussions from social media platforms into more formal academic forums.

Going into the details of our proposals, we expect that our author feedback system would not be greeted by many reviewers. We understand that introducing new regulations, especially where none previously existed, can be met with hesitation. We hope to persuade that our proposal is not to penalize the vast majority of reviewers who perform their duties diligently, but rather to establish safeguards protecting authors from the small minority of reviewers who may not meet the necessary qualifications for effective peer review. Currently, there are not many official safeguards for authors to rely on, instead of simply noting the area chairs (a process that burdens both the authors and area chairs).

Our proposal on reviewer incentives, using digital badges and reviewer impact scores, may seem very idealistic at first glance. Our incentive system is not designed to lure reviewers with a "I must get this badge!" mentality. Instead, we want to change the perspective towards peer reviewing itself. While many researchers initially engage in reviewing papers because it helps them identify novel ideas and ultimately improves their own writing, this motivation often declines over time as reviewing increasingly becomes perceived as a burden. Our goal is to provide both short-term and long-term goals to motivate reviewers to perceive reviewing as a more formal academic service, which can be used to advance their academic careers. We believe that such small changes can gradually shift the perspective towards peer reviewing for the whole academia, and eventually lead to more constructive discussions of novel ideas.

Ultimately, we hope that our proposals could help the conference build a more sustainable peer review system. Our goal is not to revolutionize the entire peer review system, but to initiate meaningful discussions about incremental changes that could enhance the peer review experience for authors, reviewers, and the broader academic community.

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

# A. Top 20 Keywords in ICLR by Year

The International Conference on Learning Representation (ICLR) is a premier academic conference in artificial intelligence, with a focus on deep learning. ICLR's open review process provides comprehensive information on submission meta data, such as the author-specified keywords, acceptance decisions, and peer reviews, making it suitable for our analysis. Here, we analyzed the author-specified keywords from the year 2018 to 2025, visualizing the top 20 keywords by year. In the analysis, we removed papers that were desk-rejected and manually matched abbreviations with their corresponding full terms (*e.g.* nlp with natural language processing). Here, Table 1 presents the annual changes of top keywords, showing both keywords that entered the top 20 and those that dropped from this ranking each year. We also provide the full list of keywords by their proportion in Figure 4 and Figure 5.

**Table 1: Top 20 Keywords In ICLR by Year**

| Year | Change | Entered Keywords | Dropped Keywords |
|------|--------|------------------|------------------|
| 2018 | - | deep learning, reinforcement learning, generative adversarial network, neural network, recurrent neural network, generative model, unsupervised learning, convolutional neural network, natural language processing, deep reinforcement learning, optimization, representation learning, adversarial examples, generalization, lstm, computer vision, transfer learning, variational inference, domain adaptation, machine learning | |
| 2019 | 3 / 20 | stochastic gradient descent, meta-learning, variational autoencoder | domain adaptation, lstm, computer vision |
| 2020 | 4 / 20 | transformer, graph neural network, robustness, interpretability | recurrent neural network, stochastic gradient descent, variational autoencoder, machine learning |
| 2021 | 3 / 20 | contrastive learning, self-supervised learning, adversarial robustness | adversarial examples, variational inference, convolutional neural network |
| 2022 | 3 / 20 | federated learning, computer vision, continual learning | unsupervised learning, deep reinforcement learning, adversarial robustness |
| 2023 | 4 / 20 | large language model, language model, offline reinforcement learning, diffusion model | meta-learning, computer vision, interpretability, generative adversarial network |
| 2024 | 2 / 20 | interpretability, computer vision | offline reinforcement learning, transfer learning |
| 2025 | 5 / 20 | foundation model, evaluation, benchmark, in-context learning, alignment | neural network, natural language processing, computer vision, generalization, contrastive learning |

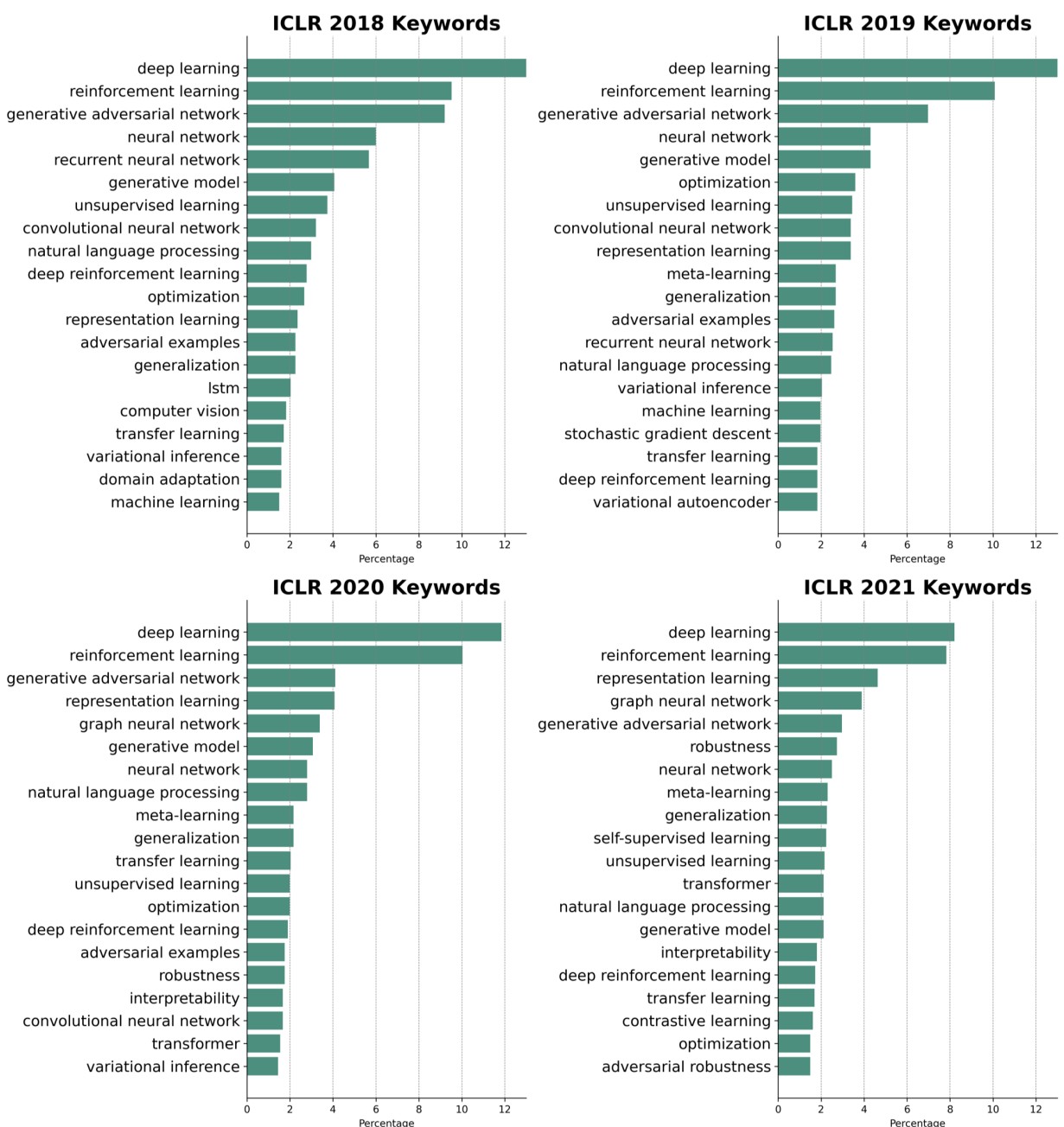

**Figure 4: Top 20 keywords from years 2018 to 2021 in ICLR Submission**

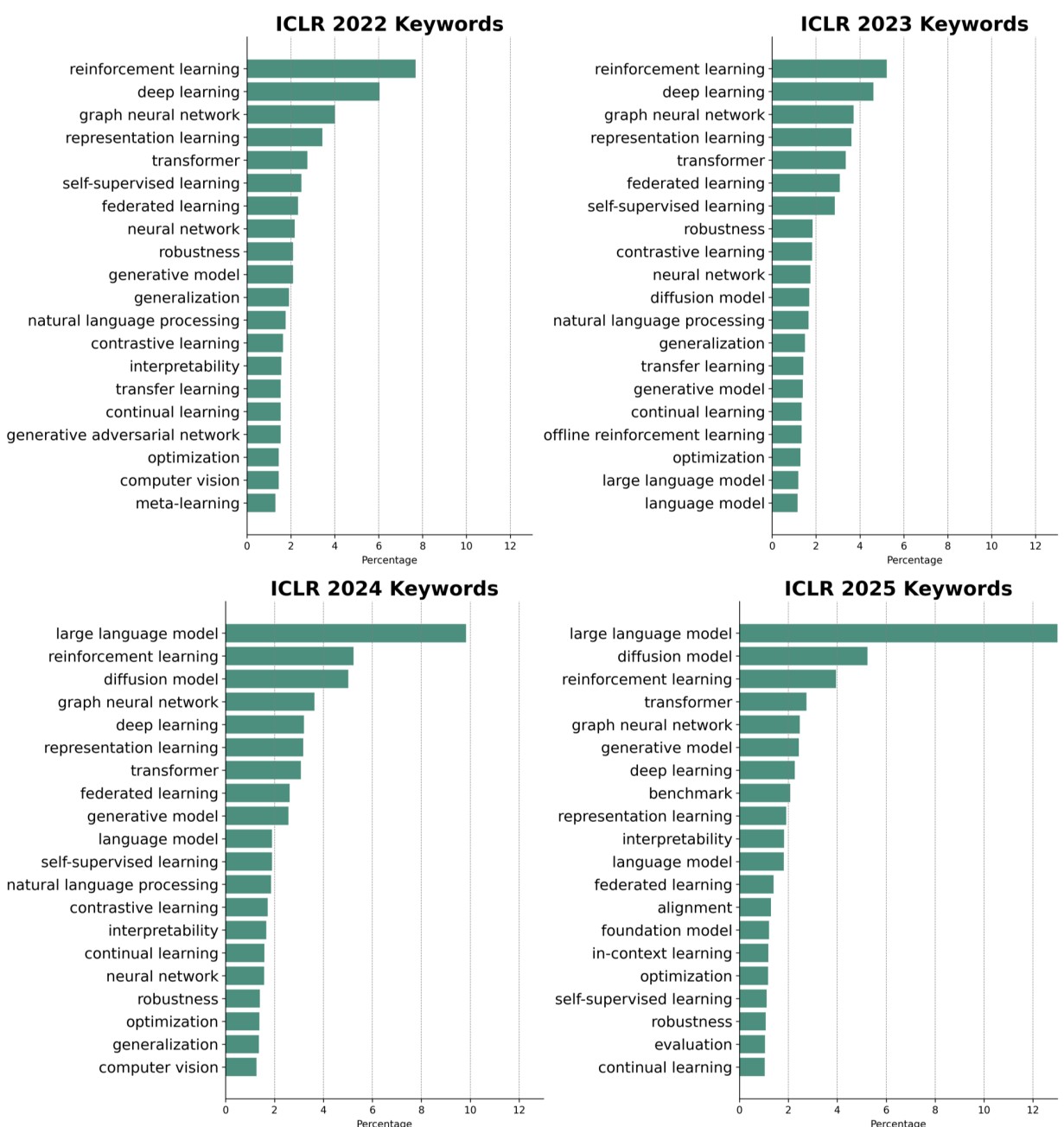

**Figure 5: Top 20 keywords from years 2022 to 2025 in ICLR Submission**

## B. LLM based Analysis of Typos.

### B.1. Why did we do the Typo Analysis?

It is extremely difficult to prove that a review is LLM-generated with current technology. One mainstream approach is to directly use LLM detection models or services. While we have tried open-sourced LLM detector models, we empirically found that these models are highly unreliable. Simply tweaking several words in the LLM-generated text can drop the confidence of these models. As such, we decided to analyze the number of typos in the peer review instead. Several works show that LLMs hardly make any spelling errors (Whittaker & Kitagishi, 2024), and as such, we considered that showing the number of spelling errors in ICLR peer reviews by year could indirectly show the effect of LLM usage in peer reviews. **We do not advocate that this typo analysis is conclusive evidence of LLM usage. There could be other factors affecting spelling error rates, such as increased usage of spell-checking tools.** Rather, this analysis serves as one of several indicators that, when combined with other observations, suggests the potential changes in review patterns.

### B.2. Analysis Method

We collected and analyzed peer reviews from ICLR 2017-2024 using the OpenReview API[8]. Due to the mathematical notations in ICLR reviews, conventional spell checkers (*e.g.* Spacy spell checker) produced excessive false positives. As such, we utilized Google's Gemini-1.5-flash LLM API for error detection. Simply, the Gemini was instructed to count the number of typos, spelling errors, and very obvious grammar errors in each peer review, using the prompts in Figure 7. In Figure 6-A, we observe a consistent decline in error rates from 2017 to 2024. The frequency of errors dropped from 7.79 per 1000 words in 2017 to 5.54 in 2024, representing a 28.8% reduction. Similarly, in Figure 6-B, the proportion of error-free reviews increased significantly from 31.4% in 2023 to 37.8% in 2024. We believe that these trends are very soft evidence of LLMs being incorporated into the peer review process, whether for generation or refinement.

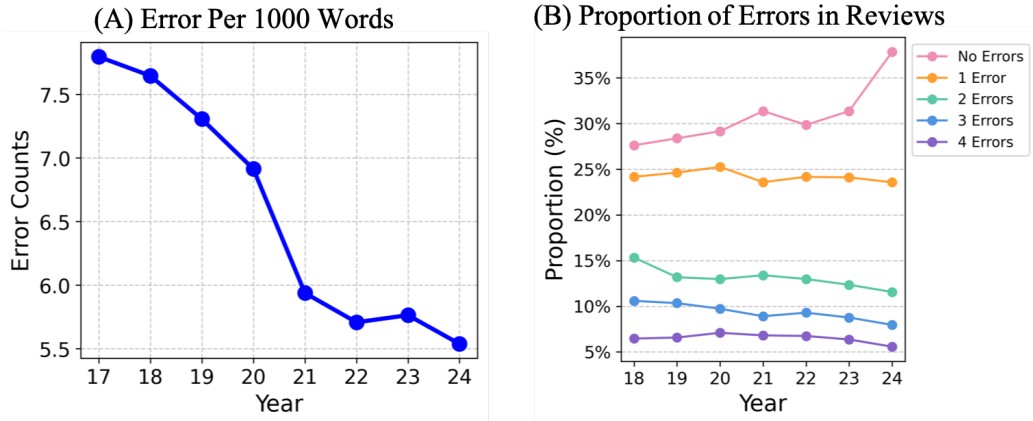

Figure 6: Error analysis using Gemini-1.5-flash

---

[8]https://docs.openreview.net/reference/api-v2

```
SYSTEM_PROMPT = """
You are a specialized spelling error detector. Your task is to analyze academic paper reviews and identify spelling
    errors and typos. Follow these specific rules:

1. Identify and classify ONLY misspelled words and typos into these categories:
    - (typo): Keyboard-related mistakes, letter swaps, missing/extra letters
    - (spell): Genuine spelling mistakes due to misunderstanding of correct spelling
    - (plural): Plural errors (only for obvious grammar errors such as plurals)

2. Do NOT identify or comment on:
    - Grammar errors (except for plurals)
    - Punctuation mistakes
    - Sentence structure issues
    - Style choices
    - Word choice appropriateness
    - Mathematical notations
    - British English vs American English

Mathematical Notation Handling:
- Ignore all mathematical expressions enclosed in markdown syntax (e.g., `$x^2$`, `$$\sum_{i=1}^n$$`)
- Common mathematical symbols and operators are not spelling errors
- Variables (like x, y, n) are considered valid when used in mathematical context
- The following notation conversions are valid and should NOT be marked as errors, or identified as spelling errors:
    'rho' to ' '
    'rho'' to '    '
    'L_1' to ' L  '
    'L_N' to ' L  '
    Similar mathematical notation conversions involving subscripts, superscripts, Greek letters, and their variants

Output Format Instructions:
Return ONLY a JSON array of error objects. Each error object must contain:
- original_word: The misspelled word
- correction: The correct spelling
- error_type: The type of error (typo, spell, or plural)

If no errors are found, return an empty array [].

DO NOT use markdown formatting or code block syntax in your response.
DO NOT include any additional text or explanations.
OUTPUT ONLY THE RAW JSON.

Example Input 1: "The quikc brown fox jmups over the lazy dog."
Example Output 1:
[{"original_word":"quikc","correction":"quick","error_type":"typo"},{"original_word":"jmups","correction":"jumps","
    error_type":"typo"}]

Example Input 2: "The value of $x^2$ and $\alpha$ shows the correlaton between variabels."
Example Output 2:
[{"original_word":"correlaton","correction":"correlation","error_type":"typo"},{"original_word":"variabels","
    correction":"variables","error_type":"spell"}]

Example Input 3: "There are important work that exist in the literature."
Example Output 3:
[{"original_word":"work","correction":"works","error_type":"plural"}]

Example Input 4: "The rho value is converted to    and L_1 becomes  L   in the equation."
Example Output 4:
[]

Important:
- Only identify spelling errors and typos
- Explicitly format the output as a JSON array of error objects
- Technical terms and proper nouns that appear in academic contexts should be verified against academic terminology
- Mathematical expressions in markdown syntax are always considered valid
- Mathematical notation conversions (Greek letters, subscripts, etc.) are valid transformations

There will only be three types of errors:
- typo: Missing letters, swapped letters, repeated letters, or obvious keyboard mistakes
- spell: Wrong letter combinations that suggest misunderstanding of the correct spelling
- plural: Plural errors (only for obvious grammar errors such as plurals)

DO NOT identify any other types of errors
"""
```

**Figure 7: Prompt used in our Analysis using Gemini-1.5-flash**

## C. Reviewer Reward System in Current AI

**Table 2: Reviewer Reward System** We analyzed the different reviewer recognition programs across major AI venues. O indicates the presence of a Reviewer Hall of Fame (RHF) program, ✗ indicates absence of RHF, and ✓ denotes venues providing in-kind compensation (IC) to reviewers. While we conducted extensive search on official conference websites, some information may be missing or may not have been publicly available at the time of this research.

| Venue | 2020 RHF / IC | 2021 RHF / IC | 2022 RHF / IC | 2023 RHF / IC | 2024 RHF / IC |
|---|---|---|---|---|---|
| AAAI | ✗ / - | ✗ / - | ✗ / - | ✗ / - | ✗ / - |
| ACL | O / - | O / - | O / - | O / ✓ | O / - |
| CVPR | O / ✓ | O / - | O / ✓ | O / - | O / - |
| ECCV | O / ✓ | | O / - | | O / - |
| EMNLP | O / - | O / - | O / - | O / - | O / - |
| ICCV | | O / - | | O / - | |
| ICDM | O / - | O / - | ✗ / - | ✗ / - | ✗ / - |
| ICLR | ✗ / - | O / ✓ | O / ✓ | O / ✓ | O / ✓ |
| ICML | O / - | O / - | O / - | O / - | O / - |
| ICRA | O / - | O / - | O / - | O / - | O / - |
| IJCAI | O / - | O / - | O / - | O / - | O / - |
| KDD | ✗ / - | ✗ / - | ✗ / - | ✗ / - | ✗ / - |
| NeurIPS | O / ✓ | O / - | O / - | O / - | O / - |
| WWW | ✗ / - | ✗ / - | ✗ / - | ✗ / - | O / - |
| Total (14) | 9 / 3 | 10 / 1 | 9 / 2 | 9 / 2 | 10 / 1 |

- AAAI (National Conference of the American Association for Artificial Intelligence)

- ACL (Association for Computational Linguistics)

- CVPR (IEEE Conference on Computer Vision and Pattern Recognition)

- ECCV (European Conference on Computer Vision)

- EMNLP (Empirical Methods in Natural Language Processing)

- ICCV (IEEE International Conference on Computer Vision)

- ICDM (IEEE International Conference on Data Mining)

- ICLR (International Conference on Learning Representations)

- ICML (International Conference on Machine Learning)

- ICRA (IEEE International Conference on Robotics and Automation)

- IJCAI (International Joint Conference on Artificial Intelligence)

- KDD (ACM International Conference on Knowledge Discovery and Data Mining)

- NeurIPS (Advances in Neural Information Processing Systems)

- WWW (International World Wide Web Conference)

**Search Method.** The AI venues were selected based on the list of CORE[9]2023's $A^*$ ranked conferences in artificial intelligence and its related fields. For the reviewer hall of fame (RHF), we primarily looked into the official venue website and searched for reviewer recognitions. To identify in-kind compensation (IC), we analyzed reviewer guidelines and calls for reviewers across venues. Notable examples include CVPR, which offers $100 payments to top reviewers, and ICLR, which has provided conference registration tickets to outstanding reviewers since 2021.

## D. Public Discourse on Review Quality

This section presents public commentary regarding current peer review practices in AI conferences. To maintain appropriate levels of privacy while preserving the authentic sentiments expressed, we have carefully rephrased certain statements using language model assistance. All quotations retain their original meaning and context while protecting individual identities where necessary. Explicit permission was obtained in cases where names are directly attributed. The views published here are provided solely for reference and have no affiliation to the specific arguments or positions presented in this paper.

*"... frustrated with the on-going review process at XX? To date, I've had: One paper desk rejected with no reason given... One review posted for the wrong paper. This was then **replaced with a review that is clearly AI generated**... Reviewers who write a few lines of text, do not provide any actionable criticism, and just give a bad score... In the last few years of reviewing for ..., I have been surprised at the number of PhD students writing reviews and the number of reviewers simply sniping other works. This needs to stop, ..."*

— **Kevin Tierney**, LinkedIn (name used with explicit consent)

*"... when you compel all authors to evaluate, that's the outcome you'll receive. None of the faculty members will have availability to assess. This results in additional students reviewing increasingly more manuscripts. I personally don't consider the evaluation system particularly valuable, to be honest. It will only **become meaningful if we begin compensating and rating evaluators.**"*

— **Annonymous**, Social Media

*"**One approach could be to enable subsequent evaluations**: scholars testing the suggested concept and methodology might remark on its effectiveness or identify flaws. In a way, this would serve as a minimal replicability verification."*

— **Annonymous**, Social Media

*"I'm also quite dissatisfied. For my XX submitted manuscripts, **not a single assessor responded to our clarifications.** It isn't unprecedented (but remains frustrating) that few reply, but absolutely nobody?"*

— **Annonymous**, Social Media

*"How can this problem be solved? **Reviewers have no incentive for making a high-quality review**"*

— **Annonymous**, Social Media

*"In the past, prestigious conferences were venues we contributed to for thorough and insightful commentary, even if it resulted in non-acceptance. Unfortunately, that incentive is more difficult to maintain currently. Nevertheless, similar to numerous colleagues, **I will ready myself for another submission round - because that remains what matters for exposure and acknowledgment.**"*

— **Annonymous**, Social Media

*"... Academia has been broken for a while, and the chief reason are perverse incentives. **You need to publish to keep funding, you need to publish to attract new funding, and you need to publish to advance your career**... It's a lot safer to invest time into creating some incremental application of a system than into more fundamental questions and approaches... "*

— **altmly**, Reddit

---

[9]https://portal.core.edu.au/conf-ranks/

