# OpenReview forum: "Position: The AI Conference Peer Review Crisis Demands Author Feedback and Reviewer Rewards"
_ICML.cc/2025/Position_Paper_Track — ICML 2025 Position Paper Track oral_

### Official Review · Reviewer_zj7j · 2025-03-12

**Significance:** 3
**Argument Clarity:** 4
**Rating:** 4
**Confidence:** 4

**Questions:**

Mostly I would like to ask about the weaknesses discussed above:

- How does desk-rejection work?
- How to address low-quality negative reviews?

**Discussion Potential:**

4

**Paper Summary:**

This paper proposed to change the ML conference peer review process by

1. Giving all papers an LLM review (to provide a point of comparison with author reviews)
2. Releasing reviews in 2 stages (summary + positives, then negatives)
3. Authors rate their reviews after stage 1 (only positives)
4. Reviewer ratings are tracked between conferences and reviewers are rewarded (eg a "digital badge" or something akin to an h-index)

Specifically, this paper advocates that their system would reduce the frequency of low-quality or LLM-generated reviews.

### update after rebuttal

The authors answered my questions. My score was already "accept" and my rating did not really change (although I didn't feel comfortable giving a "strong accept" since the work does have some weaknesses)

**Position:**

Yes

**Position In Title:**

Yes

**Related Work:**

3

**Strengths And Weaknesses:**

**Strengths**: The paper proposes a reasonable system and supports its claims about peer review with as much evidence as can reasonably be expected, given that "declining review quality" is very difficult (if not impossible) to assess objectively. The system it proposes is certainly not perfect, but is reasonable, and the authors have clearly thought through the main potential problems with it. It does seem like it could improve certain aspects of peer review. Alternative views are clearly stated.

**Weaknesses**: the main weaknesses of the paper are just the weaknesses of the proposed peer review system.

1. What happens for very low-quality papers with few (if any) positives? The authors suggest "desk rejection", but this isn't really part of the proposal and doesn't occur in existing peer review systems. A case which comes to mind is a paper that proposes a method very similar to an existing work. Normally I would write a review like "Pro: reasonable method, con: basically identical to paper X". If this was split into 2 parts (positive and negative) it might be hard to write a high-quality review. _This could potentially be mitigated by only judging based on the summary_
2. Review quality is only based on the positive, but a lot of low-quality reviews are low-quality because of the negatives. Things like "not SOTA so reject" or "this is not novel" (without providing citations) are common features of bad/lazy reviews. This system prevents the authors from giving feedback on this.
3. Digital badge systems might not be much different than existing "top reviewer" lists. Possible suggestion: "shame badges" like "repeated lazy reviewer" 🙃

**Support:**

3

---

> ### Author Rebuttal · Authors · 2025-03-29
>
> We thank the reviewer for the thoughtful feedback and for thinking through this problem with us. We're particularly grateful that the reviewer has noticed our efforts to anticipate and think through the potential problems with our proposed solution. Here, we have prepared a detailed response to each of the comments.
>
> **Q1. How does desk rejection work?**
>
> We believe that desk rejection should function similarly to current practices at major AI conferences (ICML, ICLR). While the exact evaluation criteria for desk rejection aren't fully public, reviewing the AC comments on ICLR desk-rejected papers [1] provides valuable insights. It appears that ACs quickly scan papers to filter out incomplete works, formatting issues, and irrelevant topics. We believe these operations could be automated using LLMs in the near future (NeurIPS 2024 has already employed LLMs to desk-reject papers missing the mandatory checklist).
>
> Regarding the concern raised in Weakness 1, we agree that such situations occur frequently. However, as noted in our position, we believe papers of this nature would consistently receive low scores from all reviewers. With 3-5 reviewers per paper, submissions with uniformly low ratings should exempt reviewers from evaluation.
>
> Additionally, after considering the reviewer's feedback, we recognize there might be value in allowing reviewers to flag papers they believe have no merit during the review process. However, we're concerned this could significantly complicate the system. Ultimately, we believe that conducting a pilot program would be the most effective approach, allowing us to refine the system based on real-world implementation and feedback. Once again, thank you for thinking deeply about this issue with us.
>
> [1] https://openreview.net/group?id=ICLR.cc/2025/Conference#tab-desk-rejected-submissions
>
> **Q2. How to address low-quality negative reviews?**
>
> We acknowledge that our proposed system does not directly address low-quality reviews. The reason we were not able to address this problem is because of the author-outcome bias [2], where authors are inclined to rate a review as helpful when the reviews recommended acceptance compared to rejection. If we allow authors to rate the negative reviews (regardless of the quality), we expect that most reviewers will become very liberal, which could lead to development inconsistent with the original purpose of scientific progress, potentially causing peer review to deviate from its intended purpose.
>
> As such, we believe that addressing low-quality negative reviews should be addressed in a conventional manner, where authors can note the ACs. However, we think there are two major differences. First, ACs can now cross-check whether the phase 1 reviews (questions, summary) align with the phase 2 reviews (criticism, scores). This enables ACs to identify cases where reviewers may have formed initial positive impressions but later provided negative evaluations without sufficient justification. Second, with our proposed accountability system, reviewers are incentivized to provide higher quality reviews overall, potentially reducing the frequency of low-quality negative reviews. We believe these indirect mechanisms, while not perfect, represent a meaningful improvement over the current system without introducing problematic incentives that could compromise scientific integrity.
>
> Moreover, as a suggestion, if a pilot study can be conducted, we should also collect an evaluation regarding phase 2, and analyze in which situations both evaluations were consistent and in which cases they were inconsistent, accumulating data that can be utilized as reviewer guidelines.
>
> [2]  Peer reviews of peer reviews: A randomized controlled trial and other experiments. (Arxiv)
>
> We appreciate the reviewer's feedback and hope we have addressed all concerns. Please let us know if there are any further points to discuss. Thank you.

---

> > ### Comment · Reviewer_zj7j · 2025-04-03
> >
> > Thanks for your response. I did not realize that reviewers would not be rated for papers with consistently low evaluations.
> >
> > In any case, I'm happy with your response and am happy to maintain my score. I liked the other reviewer's suggestion of comparing your approach to other forms of peer review reform that have been proposed elsewhere (including social media, since a lot of ideas aren't formally published).

---

> > > ### Author Response · Authors · 2025-04-04
> > >
> > > We thank the reviewer for their kind words and the thoughtful evaluation of our work. As the reviewer suggested, we will incorporate proposals made in other forms of media into our updated manuscript.
> > >
> > > Thank you once again for your thoughtful engagement with our work and for maintaining the scores to acceptance.
> > >
> > > Best,
> > >
> > > Authors.

---

### Official Review · Reviewer_3PNV · 2025-03-14

**Significance:** 3
**Argument Clarity:** 3
**Rating:** 3
**Confidence:** 3

**Questions:**

As mentioned in the weakness section, real-world pilot tests or data on how these proposals might work at scale would be helpful.
Additionally, more discussion is needed on how author feedback might lead reviewers to be overly positive and cautious in their reviews. It is unclear how limiting the length of the review would help.

**Discussion Potential:**

3

**Paper Summary:**

This paper starts with stating that the current peer review process for AI conferences is quickly becoming unsustainable, due to soaring submission counts, inconsistent review quality, and reviewer burnout. To remedy this, the authors propose transforming the traditional one-way system into a two-stage “bi-directional” review framework in which authors can evaluate the helpfulness of reviews before reviewers finalize their ratings. This design helps protect authors from superficial or automated (LLM-generated) feedback and motivates reviewers to provide higher-quality, thoughtful commentary. Additionally, the paper calls for a “reviewer reward system” that includes visible digital badges and metrics (e.g., “reviewer impact scores”) to track and recognize high-quality reviewing as a legitimate academic contribution. The authors suggest that these small changes could improve AI conferences to have a fairer, more accountable, and more sustainable peer review ecosystem.

**Position:**

Yes

**Position In Title:**

Yes

**Related Work:**

3

**Strengths And Weaknesses:**

Strengths
• Highlights the surge in submissions and reviewer burnout, framing them as urgent problems.
• Proposes small yet practical reforms (two-stage reviews and author feedback) that can be integrated into existing processes.
• Recognizes the emerging impact of LLMs in peer review and suggests a modest way to address them.

Weaknesses
• Lacks real-world pilot tests or data on how these proposals might work at scale.
• Author feedback could lead to reviewers to write overly positive reviews. The paper suggests limiting the length of the review, but it is unclear whether this approach is sufficient .

**Support:**

2

---

> ### Author Rebuttal · Authors · 2025-03-28
>
> We sincerely thank the reviewer for their thoughtful and constructive feedback on our work. We particularly appreciate the recognition that our paper addresses an urgent challenge facing the AI research community. We are also encouraged by your assessment that our proposed reforms represent practical solutions to peer review concerns. Here, we have provided detailed responses.
>
> **Q1. The need for real-world pilot tests or data on how these proposals might work at scale would be helpful.**
>
> We strongly agree with the reviewer's point that real-world pilot testing is essential. As we explicitly acknowledged in Section 5 (Discussions), such pilot programs should precede any widespread adoption of our proposed framework. The absence of a pilot test in our position paper is due to a practical limitation: implementing such programs requires formal conference support and infrastructure that individual researchers cannot access independently. However, we view our position paper as a necessary first step in a progression described below:
>
> 1. Currently, major discussions about peer review sustainability are mostly occurring in informal channels (Reddit, Twitter, LinkedIn)
> 2. Our position paper aims to bring these discussions to academic forums where they can receive proper scholarly attention
> 3. This academic engagement can then catalyze conference-supported pilot programs to generate the empirical data that we need to make the conference peer review system sustainable.
>
> By formalizing these proposals in an academic context, we believe our position paper could serve as a connector between informal discussions and the evidence-based implementation the reviewer and we both recognize as necessary.
>
> **Q2. Discussion is needed on how author feedback might lead reviewers to be overly positive and cautious in their reviews. It is unclear how limiting the length of the review would help**
>
> We acknowledge that this was one of the major concerns that we had when we first came up with our suggestions regarding the need for authors to evaluate the reviews. However, with carefully designed counterbalance measures, we believe such problems could be alleviated.
>
> 1. The evaluation criteria of review by authors should focus on the review quality (engagement with core details, thoroughness), rather than simply whether reviews are positives or negatives. This could potentially enable reviewers to focus on the quality of the review rather than being overly positive.
> 2. As outlined in **Section 3**, reviewers can flag submissions that fail to meet conference standards. When multiple reviewers raise such concerns, Area Chairs can initiate desk-rejection processes. Additionally, papers receiving ratings below a defined threshold would exempt reviewers from the author feedback mechanism, preventing the system from protecting substandard work. These measures could help reviewers avoid writing unnecessarily positive and cautious reviews.
> 3. Regarding the length limitation, our intention was to discourage reviewers from writing unnecessarily verbose reviews that create reviewer burden without adding value. This promotes concise reviews while respecting reviewer time constraints.
>
> Furthermore, we believe that there is a need to investigate whether similar issues exist in conferences that currently present Best Reviewer awards. It's worth analyzing whether reviewers receive these awards because they write many positive reviews or because of the qualitative aspects of their reviews. Major AI conferences such as ICML, NeurIPS, and ICLR already have Best Reviewer selection processes, which are primarily based on overall review quality, timeliness, and constructive feedback. According to our research, these awards place greater value on thoroughness and fairness rather than the positivity of reviews. Therefore, if we apply similar criteria when integrating an author feedback system, we can encourage reviewers to provide high-quality feedback rather than simply writing positive reviews.
>
> We hope that we have addressed the reviewer's concerns. Please let us know if there are additional points that need to be discussed. Thank you.

---

### Official Review · Reviewer_5DY3 · 2025-03-14

**Significance:** 4
**Argument Clarity:** 4
**Rating:** 3
**Confidence:** 5

**Questions:**

The discussion on LLMs feels somewhat superficial. Could the authors elaborate on specific strategies or guidelines for integrating LLM-generated reviews into the peer review process?

**Discussion Potential:**

3

**Paper Summary:**

This paper highlights the challenges faced by AI conference peer review systems due to the increasing number of submissions and the declining quality of reviews. It proposes a bi-directional feedback system and a reviewer rewards mechanism to mitigate the impact of non-serious reviewers. The authors advocate for empowering authors and rewarding reviewers, which helps to reduce the influence of non-serious reviewers.

**Position:**

Yes

**Position In Title:**

No

**Related Work:**

2

**Strengths And Weaknesses:**

Strengths:

1. The paper addresses a critical and urgent issue in the AI community, making it highly relevant and timely.
2. The paper effectively cites relevant data, which strongly supports its claims and enhances its credibility.
3. The paper thoughtfully discusses the potential limitations of the proposed method, demonstrating a balanced and reflective approach.
4. The introduction of LLMs is an effective suggestion.

Weaknesses:

1. The proposed method appears somewhat impractical, raising concerns about its feasibility and real-world applicability.
2. The discussion on the usage of LLMs is somewhat superficial and could benefit from more detailed strategies or practical implementation guidelines.

**Support:**

3

---

> ### Author Rebuttal · Authors · 2025-03-28
>
> We thank the reviewer for providing a positive review of our work. The reviewer has highlighted that our position has addressed a timely and urgent topic in the AI community, as well as our paper effectively delivering our position backed with relevant data. Here, we have prepared a detailed response to each of the questions and concerns raised by the reviewer.
>
> **W1. The proposed method has concerns over real-world applicability.** We respectfully disagree that our proposed method is impractical, as our method is a minimal yet actionable change that can be made with the current reviewing policy in major AI conferences.
>
> **Integration with Existing Timelines** Our proposal works within the current review schedules used by major conferences. LLM reviews can be generated concurrently with human reviewer assessments, and phased review disclosure occurs immediately after authors evaluate first-phase reviews, introducing no delays to conference timelines.
>
> **Cost-Effective Suggestion** The core components of our proposal (digital badges, phased release of reviews) require negligible monetary investment. Also, as **Reviewer h1ur** helpfully pointed out, phased review release functionality is already implemented in OpenReview's system, demonstrating the technical feasibility of our approach.
>
> **Addressing the Critical Need** The peer review system faces widespread acknowledgment of its limitations. Our finding indicates strong community support for reform, suggesting that the modest changes we propose represent a balanced approach between maintaining review quality and implementing practical, achievable improvements to the current system.
>
> While slight adjustments may be needed for practical implementations, we still believe that our proposals are highly feasible and can be incorporated in real-world scenarios.
>
> **W2. Discussion on LLMs can be improved**
>
> Thank you for raising this important question. While the minor details may be subject to further discussion, we thought about the following framework.
>
> **Reviewer Guidelines and Accountability**
>
> - Reviewers will be explicitly informed that: (1) LLM-generated reviews will be provided alongside their human reviews as a comparison point, (2) The specific LLM models and prompting strategies will remain confidential to prevent gaming the system, and (3) Authors will have a formal mechanism to flag potentially LLM-generated or otherwise irresponsible reviews.
>
> - When flags are raised, Area Chairs and Program Chairs will conduct a thorough oversight assessment following established protocols [1]. If reviewers are determined to have submitted LLM-generated or otherwise irresponsible reviews (for several works they have reviewed), substantial consequences will apply: any of their submitted papers will be rejected regardless of prior acceptance status, and they will be barred from participating in the conference (and future participation).
>
> - This approach builds on precedent already established by CVPR 2025, which desk-rejected 19 papers authored by reviewers who violated similar review integrity standards.
>
>   [1] https://cvpr.thecvf.com/Conferences/2025/ReviewerGuidelines
>
> **Usage by Authors**
>
> - Authors will receive both anonymous human-written reviews and clearly labeled LLM-generated reviews (with the AI source explicitly identified).
> - Authors can now have a reference they can leverage to raise flags. The current policy (without any LLM reviews), makes it hard for authors to challenge such irresponsible reviews.
>
> **Usage by ACs, Conferences**
>
> - Conference organizers will generate LLM reviews concurrent with the human review period, using confidentially selected AI models whose specifics remain undisclosed to prevent manipulation.
> - Conferences should establish guidelines regarding the oversight process.
> - The conference will systematically track and analyze author-submitted flags regarding suspected LLM usage, establishing a data-driven framework to identify patterns of review misconduct and continuously improve the sustainability of the peer review process.
>
> We hope that our reply has answered the concerns raised by the reviewer. If there are any additional concerns, please let us know. Thank you once again for your time and dedication.

---

### Official Review · Reviewer_h1ur · 2025-03-14

**Significance:** 3
**Argument Clarity:** 4
**Rating:** 3
**Confidence:** 5

**Questions:**

- Have such proposals been made publicly by folks on social media or in other known forums? It would be nice to document such opinions that have been experimented with or floated to others.
- Are you worried about the gamification of the system? Could reviewers now optimize for glory (through badges and rewards) without sufficiently contributing to the conference's standards by say being very liberal in their reviews of the research being published. Does it just shift the burden onto the area chairs?

Feel free to answer other questions raised in the weaknesses section.

**Discussion Potential:**

4

**Paper Summary:**

The paper addresses the challenge in the peer review system for AI conferences due to the growing number of submissions to each conference that lead to issues relating to review quality, noise in the decision making system, reviewer load/responsibilities, conference standards, etc. First, the authors propose a bi-directional blind reviewing setup where the authors get a chance to rate reviews/reviewers. The proposal includes phased rollout/revelation of reviews to the authors (hiding the weaknesses and decision recommendation section of the review in the first phase) so as to avoid retaliatory ratings against the reviewers. Second, the paper proposes a systematic reward system including digital badges and impact tracking to incentivize quality reviewing.

**Position:**

Yes

**Position In Title:**

Yes

**Related Work:**

3

**Strengths And Weaknesses:**

Strengths:
- The arguments made in the paper are well supported by observations from published works as well as anecdotal evidence around the status of peer reviewing in our field.
- The proposed solution for the phased rollout does help with withholding the retaliatory ratings from authors to reviewers. This is a technique deployed by Airbnb (for example) to let both parties rate each other 1 to 5.
- The study around the sudden reduction in the number of errors in reviews is interesting, and does demonstrate the growing use of LLMs or other writing tools to assist with reviewing.

Weaknesses:
- The impact of growing number of submissions on the reviewers is not well studied or analyzed in the paper. The number of submissions is public information but so is number of reviewers, so the average load per reviewer can be noted from that. A small study could reveal how much time reviewers give to one paper, whether they read it fully (for example reviews mentioning that they did not check the proofs or see the appendix), confidence scores in the reviews etc. could be studied.
- On similar lines, while authors present information about reviewer awards and incentives, there is no good study around a reviewer's motivation to do reviewing. There should be a tradeoff between burden on the reviewers and the quality of decisions that can be made using their reviews. If there are a 100 fields including 50 scores to be filled in to enter a review, the burden on the reviewer is maximized even if it makes sure that the review is good quality.
- I am not sure if the thoughtfulness or insightfulness of the questions asked to the authors in a review tell anything about whether a review is good or not. Even if it does, I wonder if it is highly correlated with the weaknesses section (which is withheld). I can agree with hiding the score in the first phase. This has been practiced in a few conferences over the past years and I wonder if results were published from those studies.
- I am also not convinced by the approach to deal with LLM reviews, but I understand it is a tough challenge in general too.
- Minor: Openreview does allow for phased revelation of scores/sections of the review over different phases of the review process, so this need not be mentioned as a limitation of the approach.

Overall, I believe the review quality is the major concern posed by this paper but the authors shy away from discussing the problem from the direction.

**Support:**

3

---

> ### Author Rebuttal · Authors · 2025-03-28
>
> We sincerely thank the reviewer for their thorough assessment of our work and for providing valuable insights. We're pleased that reviewer h1ur has found our position well-supported by observations and our analysis interesting. Below, we provide detailed responses.
>
> **Q1.Similar proposals have been made?**
>
> Yes, similar proposals regarding bidirectional evaluation between reviewers and authors have been discussed in forums such as Reddit, LinkedIn, Medium, etc. However, to the best of our knowledge, our contributions of using LLM reviews as a quality control (deterring reviewers from solely relying on LLMs), Digital Badges, and implementing phased reviews are novel elements. While fragmented components of our ideas may exist in public forums and research papers, we haven't seen these proposed together as a combined perspective. Following the reviewer's suggestion, we would be happy to include them in our paper writing in our revised manuscript.
>
> **Q2. Potential Gaming**
>
> We recognize that while Digital Badges leverage positive gamification elements (participation, motivation), they could potentially incentivize reviewers to optimize for rewards through overly liberal reviewing rather than maintaining review standards. To address these concerns, we believe we should use carefully designed evaluation metrics that reward quality over leniency. Badges should be provided to reviewers who demonstrate thoroughness, identification of crucial issues, and constructive feedback, but not simply through positivity of the assessments.
>
> Regarding AC burden, we believe our proposals can reduce rather than increase their workload. The badge system provides AC with automated reviewer performance, making it easier to identify both high-performing and those who may be gaming. This actually streamlines their oversight process rather than adding to it.
>
> **W1-2. Analysis on the Load on reviewers**
>
> As the reviewer suggested, we analyzed the number of reviews and the total number of reviewers in ICLR 2023-2025, as data for the total number of reviewers was only available for this period.
> | Year | # Review | # Reviewers | Reviews/ Reviewer |
> | -- | - | - | - |
> | 23| 14,351| 5,734| 2.50 |
> | 24| 28,028| 8,950| 3.13|
> | 25| 46,747| 15,249| 3.07|
>
> While the reviews per reviewer have increased over the past two years, we cannot conclusively attribute degraded review quality solely to increased reviewer load. We acknowledge that these metrics alone do not fully quantify the complexity of tasks reviewers must complete. As such, a comprehensive examination of reviewer workload would indeed be valuable future work. However, we hope the reviewer understands that such analysis itself constitutes substantial research content that, unfortunately we could only address briefly in the current study
>
> **W3. Correlation between Q.A and review quality.** We appreciate this thoughtful concern. While we understand this perspective, our experience suggests there may be a meaningful relationship between the two.
>
> In our observations, when reviewers pose insightful questions, it often indicates they've engaged carefully with the paper. To ask something truly relevant or thought-provoking typically requires understanding the work's core contributions. From our experiences on both sides of the review process, authors can sense when questions reflect genuine engagement with their work versus more surface-level reading. This suggests that question quality might serve as one helpful indicator. If reviewers know their questions contribute to how their reviews are evaluated, this could encourage thorough reading of papers. And once someone has taken the time to understand a paper deeply, they're more likely to provide thoughtful feedback throughout their review. However, we certainly agree this approach would need careful monitoring over time.
>
> **W4. LLM review.** Thank you for your understanding. Here, our proposal doesn't claim to entirely solve this LLM-review problem but rather introduces a way that may psychologically discourage reviewers from exclusively using LLMs while simultaneously providing authors with a reference point to identify potentially LLM-generated reviews. Currently, authors who suspect receiving fully LLM-generated reviews face significant barriers to challenging such reviews or reporting them to AC. By providing an official LLM-generated review as a baseline comparison, our approach gives authors a soft (and official) reference that can help identify suspicious reviews and provide legitimate grounds for raising concerns. This creates an additional accountability layer that could discourage the submission of low-effort, fully automated reviews while empowering authors with a means to advocate for a fair evaluation of their work.
>
> **W5. Minors.** Thank you for sharing! We will modify the Practical Challenges section to accommodate more meaningful limitations. We believe that the potential of Gamification could be thoroughly discussed.

---

### Decision · Program_Chairs · 2025-04-30

**Decision:**

Accept (oral)

**Comment:**

Reviewers felt this was a comprehensive and thorough proposal regarding how to handle the review process at major ML conferences. The work was found to be in-depth and details in terms of possible pitfalls and how to deal with them. Additional points were raised during the review process, but nothing super critical. Some reviewers commented on lack of real-world tests, which I don't think is necessary for a proposal such as this (that would be the next natural step). Other weaknesses were more like suggestions on the proposal, which I view as a positive sign that this work will spark discussion in the community and at the conference.

Authors should revise the title to state the position, as instructed in the CFP.  "Towards..." is a direction, not a position.  It may help to think of a verb to include.